# Synthetic Iowaite Can Effectively Remove Inorganic Arsenic from Marine Extract

**DOI:** 10.3390/molecules26103052

**Published:** 2021-05-20

**Authors:** Jing Ji, Wenwen Huang, Lingchong Wang, Lu Chen, Yuanqing Wei, Rui Liu, Jianming Cheng, Hao Wu

**Affiliations:** 1College of Pharmacology, Nanjing University of Chinese Medicine, Nanjing 210023, China; jier4522@163.com (J.J.); 993wlc@njucm.edu.cn (L.W.); chenlu9511@163.com (L.C.); wei_yuanqing@njucm.edu.cn (Y.W.); liurui@njucm.edu.cn (R.L.); 2Jiangsu Key Laboratory of Research and Development in Marnie Bio-resource Pharmaceutics, Nanjing 201123, China; 3Jiangsu Collaborative Innovation Center of Chinese Medicinal Resources Industrialization, Nanjing 210023, China; 4Third Institute of Oceanography, Ministry of Natural Resources, Xiamen 361005, China; wwhuang@tio.org.cn

**Keywords:** iowaite, As(III), As(V), remove, mactra veneriformis

## Abstract

For the removal of arsenic from marine products, iowaite was prepared and investigated to determine the optimal adsorption process of arsenic. Different chemical forms of arsenic (As(III), As(V)) with varying concentrations (0.15, 1.5, 5, 10, 15, and 20 mg/L) under various conditions including pH (3, 5, 7, 9, 11) and contact time (1, 2, 5, 10, 15, 30, 60, 120, 180 min) were exposed to iowaite. Adsorption isotherms and metal ions kinetic modeling onto the adsorbent were determined based on Langmuir, Freundlich, first- and second-order kinetic models. The adsorption onto iowaite varied depending on the conditions. The adsorption rates of standard solution, As(III) and As(V) exceeded 95% under proper conditions, while high complexity was noted with marine samples. As(III) and As(V) from *Mactra veneriformis* extraction all decreased when exposed to iowaite. The inclusion morphology and interconversion of organic arsenic limit adsorption. Iowaite can be efficiently used for inorganic arsenic removal from wastewater and different marine food products, which maybe other adsorbent or further performance of iowaite needs to be investigated for organic arsenic.

## 1. Introduction

Besides global warming, industrialization and urbanization are the major causes for severe environmental threats, such as air, soil, and water pollution, which resulted in the rise of hazardous heavy metal contents in the environment [1,2]. Arsenic naturally presented in soil can be mobilized and transported, leading to its high concentrations in aquifers, which are sources of drinking water and seawater [3]. Arsenic pollution in aquatic environments is of great global concern. Toxicant arsenic exposure may lead to disturbances of essential metal and protein metabolism in organisms. Accumulation of arsenic in benthic invertebrates may be passed up the food chain and result in their continued persistence in the environment [4]. At present, arsenic is generally found in rock, soil, water, air, plant, and animal tissues. It exists in four valency states: −3, 0, +3, and +5. However, prevalence of arsenic is concerned in its toxicity, which depends on its chemical form [5]. In general, As(III) types are the most toxic in all formats and followed by As(V) and organic types sorts [5,6]. Arsenic is mobilized through a combination of natural processes such as weathering reactions, biological activity, and volcanic emissions. Most environmental arsenic problems are the result of mobilization under natural conditions. The most natural source of arsenic is volcanic and geological activities. It evaporates at low temperature and reaches atmospheres and other media. Natural arsenic sources are larger than anthropogenic sources, and their ratio is around 40–60% [7]. The earthquake that occurred in Canakkale in 2013 triggered the increase of arsenic in underground waters, resulting in mobility in the fault line [8]. However, humans have also contributed to rising arsenic levels through mining activities, combustion of fossil fuels, and the use of arsenical pesticides, herbicides, crop desiccants, and additives to livestock feed, particularly for poultry. Although the use of arsenical products such as pesticides and herbicides has decreased significantly in the last few decades, their use for wood preservation is still common. A range of uncontrolled anthropogenic activities in the environment due to the use of arsenical compounds, at least locally, will remain for some years. Due to the various sources of arsenic in the environment, drinking water and seawater probably pose the greatest threats to human health. From the standpoint of diversity, it is a long way in studying arsenic sorption from sea water and/or marine products. The rate at which sorption occurs is of the utmost importance when designing batch sorption systems. As a result, time-dependence of such systems should be established under various process conditions.

The toxicity of As(III) is about 60 times more than that of As(V) due to its higher cellular uptake [9]. Arsenic might cause neurological damage when its concentration is slightly higher than 0.1 mg/L, and a concentration of 0.2 mg/L produces dermatosis [10,11]. For polluted seawater, arsenic is typically dominated by As(V), although some As(III) is invariably present and is of increasing importance in anoxic bottom waters. Ratios of As(V)/As(III) are typically in the range of 10–100 in open seawater [12]. Inorganic arsenic (InAs) has long been considered to be a teratogen in multiple mammalian types [13,14]. The World Health Organization (WHO) suggested an acceptable concentration of 0.01 mg/L based on the potential health risk and the practical quantitation limit. Classical technologies applied for the removal of toxic metals from aqueous solutions, such as ion exchange, chemical precipitation, membrane processing, and sorption, are often inefficient or expensive, especially when heavy metals are present at low concentrations [15,16]. Most methods have both advantages and drawbacks, which might cause great interest in the use of other sorbent materials. Many researchers have used nanoparticles as adsorbents to remove water pollutants, including arsenic, through modifying the properties of nanoparticles in improving reactivity, biocompatibility, stability, charge density, multi-functionalities, and dispersibility. For arsenic removal, nanoadsorbents have emerged as potential alternatives to existing conventional technologies. Layered double hydroxides (LDHs), a family of inorganic layered materials, have recently attracted increasing attention because of their potential applications in a wide range of areas such as catalysis, sorption, and nanocomposite and drug delivery [17,18,19]. Iowaite, an Mg/Fe-based LDH intercalated with chloride, was synthesized to evaluate its performance for arsenic removal from water and shellfish.

## 2. Materials and Methods

### 2.1. Reagents and Instrumentation

All standards As^3+^ (GBW08666), Monomethylarsonic acid (MMA, GBW08668), Arsenobetaine (AsB, GBW08670), AsC (GBW08671), DMA (GBW08669) and As^5+^ (GBW08667) were obtained from the Chinese Academy of Metrology. NaAsO_2_ (>90%, SLBZ4530) was obtained from ALDRICH Chemistry- (Sigma-Aldrich, St. Louis, MO, USA). These As species stock solutions were diluted with deionized water (Milli-Q 18.2 MΩ cm) to the desired concentrations before use. For the analytical procedures, the following chemicals were used: di-ammonium hydrogen phosphate from Sigma-Aldrich, and hydrogen peroxide and nitric acid from Merck. Total As standard of 10,000 μg/mL from CPI International was used for making standards for total As determination. Indium standard of 1000 μg/mL from CPI International was used to prepare the internal standard for total As. Other analytical reagents (MgCl_2_·6H_2_O, FeCl_3_·6H_2_O and NaOH) were all from Sinopharm Chemical Reagent Co., Ltd. Beijing, China.

Different types of iowaite were screened for subsequent experimental study to determine the best sorption effect through sorption studies using Magnetic Stirrer (DF-101Z, Airlink, Shanghai, China). The iowaite samples and those samples that reacted with arsenic-bearing solutions were characterized through a variety of techniques, including powder X-ray powder diffraction (Bruker D8, Cu Kα radiation), scanning electron microscopy (Hitachi TM3000) and X-ray Energy Dispersive Spectrometers (Oxford X-act, Oxford, England, UK), Fourier transform infrared spectroscopy (FT-IR, Nicolet iS5, Thermo Fisher, Madison, WI, USA). All species of arsenic were analysis by High-Performance Liquid Chromatography Inductively Coupled Plasma Mass Spectrometry (HPLC-ICP-MS) (Nex ION 350, Perkin Elmer, Waltham, ME, USA). Content of total arsenic was determined by ICP-MS. The instrumental operating condition for HPLC-ICP-MS is shown in Table 1.

### 2.2. Preparation of Iowaite

The conventional coprecipitation method, as reported elsewhere [20], was used to synthesize iowaite in this study. The Mg/Fe molar ratio of synthetic iowaite was controlled by using initial salt solutions. Salt solutions were prepared by mixing 2.5 mol/L magnesium chloride and 1.0 mol/L ferric chloride, which were added dropwise into a large beaker containing sodium hydroxide solution under substantial stirring. The solid products were purified by dialysis with Nanopure water until the conductivity of the rinsing water was lower than 0.1 ms/cm. Finally, the solids were dried for arsenic removal experiments. A dialysis bag was employed instead of centrifugation in the preparation procedure, which eliminates the tedious operation of repeated centrifugation and ultrasonic for evacuation structure in the later stage. Freeze-drying and oven-drying with different temperatures were employed in the drying process. Samples were prepared by dilution from stock solutions and solid. The volume of each sample was 10 mL, but it was 50 mL in the pH experiments.

### 2.3. Materials Modifactions

A similar type of modification is also possible with other types of LDHs (layered double hydroxide). One of the differences is in the choice of metal atoms in the surface. By contrast, in the LDH structure, some of the divalent cations of the brucite-like sheets are isomorphously replaced by a trivalent cation. The materials of MgFe-SO_4_-LDH, MgAl-SO_4_-LDH, and FeAl-SO_4_-LDH were prepared similarly. The sorption process was the same as iowaite. Modify groups of -OH, PEI and HS- were all prepared for further use.

## 3. Batch Sorption Experiments

All the arsenic removal experiments were performed via the batch sorption technique by reacting 10 mL of standard solution of H_3_AsO_4_ or NaAsO_2_ of various concentrations with 10 mg of synthetic iowaite. The stock solutions were prepared in deionized water at the initial concentration of 1000.0 mg/L. For different pH studies, 50 mL polyethylene centrifuge tubes were used. The pH of the solutions was adjusted to 3–11 with NaOH and HCl solutions. The mixtures were then mixed using DF-101Z magnetism Stirrer with constant temperature heating.

### 3.1. Influence of pH

The effect of pH on arsenic removal was determined by adding 10 mg of iowaite in a medium containing arsenic (NaAsO_2_ and H_3_AsO_4_) with the pH ranging from 3 to 11. The initial solution was adjusted to the desired pH with diluted or concentrated HCl and NaOH solutions before being mixed with the sorbent. The change in the working volume due to the addition of HCl and NaOH calculated. The equilibrium and kinetics experiments were performed at an unaltered initial pH (NaAsO_2_ pH 7 and H_3_AsO_4_ pH 3).

### 3.2. Kinetics of Sorption

Specifically, the kinetics of arsenic sorption were investigated by reacting iowaite with 0.15, 1.5, 5, 10, 15, and 20 mg/L arsenic solutions (H_3_AsO_4_ and NaAsO_2_), and the isotherm sorption experiments were conducted using the same initial arsenic concentrations. During the kinetics experiments, the sample bottles were sealed and placed in a constant-temperature water bath shaker at 25 °C for contact periods of 5 s, 30 s, 1 min, 2 min, 15 min, 30 min, 60 min, 120 min, and 240 min. After the sorption experiments, the solution samples were decanted and filtered through a 0.02 μm microporous membrane (for HAsO_3_ and NaAsO_2_, respectively). On the basis of the initial arsenic concentration *C*_0_ (mg/L) and the equilibrium concentration *C_e_* (mg/L), the sorption arsenic capacity *Q_e_* (mg/g) and the removal rate *R* (%) were calculated by Formulas (1) and (2). *V* is the volume (L) of the arsenic solution, and *M* is the mass (mg) of the adsorbent.
(1)$$Q_{e}=\frac{\left(C_{0}-C_{e}\right)\cdot V}{m}\;R=\frac{C_{0}-C_{e}}{c_{0}}\times 100\%$$

All the experiments were carried out under ambient conditions at 25 °C, except for the thermodynamic studies at 25 °C, 45 °C, 65 °C, and 85 °C for a contact period of 3 h. At the end of each mixing period, the biomass was removed by filtration through a 0.22 μm microporous membrane filter, and the filtrate was analyzed by ICP. All the experiments were conducted in triplicate, and the averages of the measurements for each treatment was used.

### 3.3. Adsorption to the Mactra veneriformis Extract

The standard solution of AsC, AsB, As^3+^, DMA, MMA, and As^5+^ was accurately weighed and diluted with distilled water to 1000 ng/mL as standard reserve solution by adding 0.1% HNO_3_. *Mactra veneriformis* is a kind of mollusk, also a high arsenic content marine product, which cultivated in the coastal area of Jiangsu Province. The extract from *Mactra veneriformis* was prepared by a member of our research group. The extraction is the supernatant after processing of water extraction and ethanol precipitation. Iowaite was added to the sample at enough of 20%, and sorption occurred in saturated aqueous solution under 85 °C for 2 h, which is the optimal sorption process vitrificated by standards. Iowaite was removed by filtration through a 0.22 μm cellulose acetate membrane filter, and the filtrate was analyzed by HPLC-ICP-MS.

## 4. Results

### 4.1. Characterization of Synthetic Iowaites

Surface morphology and elemental composition of iowaite before and after adsorption were analyzed by SEM and EDS. All scale ranges among 1–100 μm were observed, but no arrangement was found. Figure 1 shows the SEM images, A is the image of iowaite, B and C are the images after adsorption. All images were observed under the scale range of 1 μm, sparse void can be seen at iowaite. After adsorption, most of the voids were filled in B and C. The change of morphology really reacts, and new chemicals are formed with different morphology. Figure 2A is the EDS of iowaite, 2B is iowaite-NaAsO_2_. From the result of EDS, different elemental compositions can list that the content of As and S increased obviously, while the proportion of O decreased. This may indicate the adsorption of As and S or the exchange of O and As. Figure 3 shows iowaite-NaAsO_2_ EDS images of each element channel, the complexation result of each element is not as obvious as Figure 3. It is certain that the arsenic has been adsorbed in some form. Figure 1 also shows that iowaite exhibited a loose arrangement, and arsenic was uniformly attached to it. The elevated temperature possibly expanded the internal space of iowaite and allowed it to firm sorption with arsenic. The Nicolet iS5 FT-IR after 40 scans within 400–4000 cm^−1^ at a resolution of 4 cm^−1^ by measuring the IR absorbance of a KBr disk that contained 1% of the sample. The O-H stretching vibration frequency of anhydrous base sodium hydroxide was 3550–3720 cm^−1^. The FT-IR spectra showed that chemical changes occurred in the synthesis process, and no trace of NaOH residues was found. The characteristic peak of iowaite was relatively obvious from the FT-IR spectra. The transmittance of 3350 and 1680 cm^−1^ decreased after sorption, which was consistent with the results of the formation of new compounds. Martemianov [21] proposed that good retention of arsenate is due to the chemisorption reaction between arsenate anion and ferric cation at the adsorbent surface, in which ferric arsenate is formed. The change trend of peak shape of FT-IR matched well with XRD. X-ray diffraction patterns were collected at a scanning rate of 2°/min from 2*θ* = 5° to 2*θ* = 65° with Co Kα radiation (*λ* = 0.17902 nm) on a Rigaku Miniflex X-ray diffractometer with a variable slit width. Diffraction peak width is related to order degree or crystallization degree; a high-order degree corresponds to a sharp peak. Through search of phase PDF card in Jade 6, Figure 2A was identified as iowaite, which chemical formula is Mg_4_Fe(OH)_10_Cl(H_2_O)_3_ with 4.4 FOM (figure of merit). Iowaite was accurately prepared and the optimization of process did not affect its major structure. After sorption, non-sharp peak can be found in Figure 2B, which suggested that Figure 2B is a molecule with low crystallinity, that may be the mixture of iowaite and adsorbed substances. Comparison of the two figures shows that the adsorption process is a combination of chemical and physical changes. One part of arsenic may chemically bind with iowaite and the other part of arsenic may just be physically adsorbed. Figure 2B may show the mixed iowaite and arsenic, in which ferric arsenate may be formed [21]. Figure 3 shows EDS images of each element channel after adsorption. Figure 3B is the channel of Mg, Figure 3C is the channel of Fe, Figure 3D is the channel of S, Figure 3E is the channel of As (which is not clear due to the color of the mark). Figure 3A covers all channels from B to E. From the trajectory distribution of each element, the integrative trend of As cannot be shown. Figure 4 was drawn with the concrete data from EDS, comparing the highest values of each orbital peak to principal lines keV, specific elements are found. The *x* axis shows the principal lines keV in Figure 4, Figure 4A is the EDS of iowaite, 4B is iowaite-NaAsO_2_. From the results of EDS, different elemental composition can list that the content of As and S increased obviously, while the proportion of O decreased. This may indicate the adsorption of As and S or the exchange of O and As. It is certain that the arsenic has been adsorbed in some form.

### 4.2. Influence of pH on the Arsenic Removing with Iowaites

The effect of pH on sorption was examined over a wide pH range. The pH of a medium is among the most important factors that can influence the sorption process. Therefore, preliminary experiments were performed to determine the optimum pH for maximizing arsenic removal, prior to investigating the kinetic and thermodynamic aspects of sorption. As a general trend, an increase in adsorbed metal amount was observed with the increase in pH. The results from Qinghai Guo [22] showed only iowaite peaks in the XRD patterns of all the reacted solid samples. Nevertheless, the mass percentage of the remaining nano-iowaite decreased from 99.6% at pH 6 to 97.0% at pH 2, thereby confirming that iowaite was slightly more unstable at acidic pH than at alkaline pH, which is similar to this study. This phenomenon resulted in a slight release of Mg, and the amount of Mg released was far less than the body’s recommended daily intake, which was verified by ICP-MS. Using edible grade MgCl_2_ as raw material ensures the safety of residues. A systematic test on the stability of iowaite at wide ranges of pH and arsenic concentration must be performed before it will be applied into practice under acidic conditions. A large range of pH was tested, which indicated a change in iowaite but no effect on the sorption of As. By contrast, pH values in the alkaline range were not preferred due to the possible precipitation of metal hydroxides (Table 2 and Table 3). Figure 5 expresses the direct results.

The experiments performed within this context for AsO_4_^3−^ and H^+^ ions at pH values of 3.0, 5.1, 7.1, 9.2 and 11.1 resulted in 99% biosorption values of different pH levels for H_3_AsO_4_. For NaAsO_2_, a decrease in arsenic removal was observed with the increase in pH (pH > 7). However, in liquid–solid sorption systems, the situation is much more complicated as the behavior of ions in solution or on solid depends on factors such as the inter-ionic forces, hydration energy, availability of sorption sites, and relative stability of sorted ions at these sites. The influence of Na^+^ ionic strength on arsenic sorption might exert some effect on ion exchange between liquid and solid, and other ions may have the same influences. When the pH exceeds 9, the sorption capacity of arsenic decreases, which is due to the competitive sorption between excessive OH^−^ and arsenic in the solution. Given that OH^−^ is electronegative, it is highly likely to produce an anion exchange with Cl^−^ between sorption material layers. To study sorption under acid and salt solution conditions, subsequent studies were carried out under the condition of no change in pH. The initial solution was diluted with distilled water before being mixed with the sorbent for further experiments as an optimum pH condition.

### 4.3. Kinetics of Sorption

Kinetic studies of sorption on iowaite were carried out at the initial metal concentrations of 0.15, 1.5, 5, 10, 15, and 20.0 mg/L for times of contact that ranged from 1 to 180 min. The experiments were conducted at 25 °C. The sorption of adsorbents on the surface of adsorbents is a dynamic process; the sorption time changes with the environmental conditions. Therefore, the sorption rate is a key factor to evaluate the properties of adsorbed materials. The interaction between iowaite and arsenic was described by an empirical formula. The pseudo-first-order kinetic model (Formula (3)) and the pseudo-second-order kinetic model (Formula (4)), which were derived by Lagergren [23,24], are expressed as follows:(2)$$\ln\left(q_{e}-q_{t}\right)=\ln q_{e}-k_{1}t$$
(3)$$\frac{t}{q_{t}}=\frac{t}{q_{e}}+\frac{1}{k_{2}q_{e}^{2}}$$
where *q_t_* is the concentration of sorbed ion on the solid at time *t* (mg·g^−1^), *q_e_* is the concentration of sorbed ion at equilibrium (mg/g), and *k*_2_ is the second-order rate constant (g·mg^−1^·min^−1^). The values of *q_t_* in the equations above were calculated using Formual (2). The kinetic results of the removal of H_3_AsO_4_ and NaAsO_2_ with iowaite adsorbent are shown in the following figure.

Figure 6 shows arsenic removal from aqueous solution with different initial concentrations using iowaite at 25 °C with the initial pH. When the initial arsenic concentration was low, iowaite was fast and took a short time to reach the equilibrium (less than 5 min), but arsenic was released back into the solution with iowaite exposure of more than 5 min. The arsenic removal ratios in aqueous solution after 5 min reached 84.5% and 81.3% for the initial NaAsO_2_ concentrations of 0.15 and 1.5 mg/L, respectively, whereas those for H_3_AsO_4_ were 90.6% and 98.1%, respectively. The arsenic solution with high initial concentration needs a longer time to reach the equilibrium state. The time required to reach equilibrium in all cases was less than 60 min. When the concentration was above 10 mg/L, both the material and AsO_4_^3−^ may be reconstructed due to increased acidity. The change in AsO_4_^3−^ under different pH values can be verified from the literature [25]. The reconstruction of iowaite took only a few minutes and did not decrease the absorption ability of arsenic.

Parameters of the pseudo-first- and second-order models are all shown in Table 4 and Figure 7. The dynamic sorption process of arsenic removal was adapted to three stages: the initial stage of sorption was mainly in addition to the adsorbent. During this period, there were multiple sorption points on the surface of iowaite; meanwhile, the adsorbent exhibited a large arsenic concentration difference on the surface and inside, so the sorption rate was slower than the initial stage. As the sorption process continued, the arsenic content in the liquid decreased, and external sorption spread inward and reached a saturation state. As a result, the sorption speed decreased due to the enhanced resistance. Subsequently, the effect mostly occurs at high energy sites during sorption, and the pressure thrust caused by the concentration difference gradually decreases, resulting in the final equilibrium state of sorption [26,27]. The pseudo-first-order kinetic model describes the sorption process controlled by the diffusion motion of boundary molecules, while the pseudo-second-order model assumes that the sorption velocity is controlled by the sorption chemical mechanism, and electron transfer and sharing occur between iowaite and arsenic.

### 4.4. Thermodynamic Fitting for the Sorption

The sorption process of adsorbent to solute is a dynamic equilibrium process. The sorption process is usually described by the sorption equilibrium isothermal equation (line). These isothermal formulas refer to the relationship between the sorption amount of solute on the adsorbent surface and the equilibrium concentration of solute in the solution at a certain temperature. The Langmuir isotherm (Formula (5)) and Freundlich isotherm (Formulas (6) and (7)) are commonly used to describe the sorption process [28]. The Langmuir isothermal sorption model assumes that absorption occurs on homogeneous surfaces through monolayer sorption; no interaction is reported between adsorbed molecules, and the binding sites are limited. The sorption capacity reaches the maximum when the adsorbent surface is saturated. The results of XRD also verified that adsorption is not only through monolayer sorption. Freundlich isothermal sorption is an empirical formula for multimolecular sorption based on the Langmuir model:(4)$$\frac{1}{q_{e}}=\frac{1}{q_{max}}+\frac{1}{k_{3}q_{max}c_{e}}$$
where $q_{e}$ is the sorption capacity at equilibrium (mg/g), $c_{e}$ is the concentration of arsenic solution at equilibrium (mg/L), $q_{max}$ is the maximum sorption capacity of adsorbent (mg/L), and $k_{3}$ is the equilibrium sorption constant of Langmuir (L/mg).
(5)$$q_{e}=k_{4}c_{e}^{1/n}$$
(6)$$\log q_{e}=\log k_{4}+\frac{1}{n}\log C_{e}$$

*k*_4_ and *n* represent factors that may affect sorption in Formula (6), and Formula (7) is its linear expression.

As shown in Table 5, the correlation coefficient (*R*^2^) of NaAsO_2_ and H_3_AsO_4_ isothermal sorption data at different temperatures fitted by the Langmuir model was high (>0.92), while the Freundlich isothermal sorption model was low (>0.46). For NaAsO_2_ and H_3_AsO_4_, the values of 1/*n* fitted in the Freundlich isotherm model were between 0.1 and 0.5 (except H_3_AsO_4_ at 25 °C), indicating that both NaAsO_2_ and H_3_AsO_4_ were favorably adsorbed by iowaite. To confirm the As(III)/As(V) sorption mechanism, the sorption thermodynamic parameters (Δ*G*^0^, Δ*H*^0^, and Δ*S*^0^) were determined from the Gibbs free energy equation (Formula (8)) [29].
(7)$$\Delta G^{0}=\Delta H^{0}-T\Delta S^{0}$$

Δ*G*^0^ is the Gibbs free energy change constant, Δ*H*^0^ is the standard enthalpy change, and Δ*S*^0^ is the standard entropy change. For every spontaneous sorption process, Δ*G*^0^ value must be negative. In sorption equilibria, the equilibrium constant *k*_3_ in the Langmuir constant is related to the Gibbs free energy change by using Formula (9) [30].
(8)$$\Delta G^{0}=-RT\ln k_{3}$$

*R* (8.314 J mol^−1^K^−1^) is the molar gas constant; *T* is the absolute temperature in Kelvin. Formulas (8) and (9) can be converted to Formula (10).
(9)$$\ln k=-\Delta H^{0}/RT+\Delta S^{0}/R$$

Formula (8) shows a linear relationship between Δ*H*^0^ and Δ*S*^0^ with the slope equal to the reference temperature. Thus, the process of sorption is the same for all compounds and occurs with a constant change in Δ*G^0^* of interaction, where a change in interaction enthalpy is offset by a change in entropy upon binding and vice versa. The thermodynamic parameters are closely related to the actual application of sorption reaction. The characterization of the changes in the internal energy of the adsorbent and the product during the reaction can reveal whether the sorption process is a spontaneous reaction. Thermodynamic parameters are presented in Table 6. In general, a negative Δ*G* value suggests that the sorption process is a spontaneous reaction, and the value indicates the sorption forces. The reaction enthalpy change Δ*H* value can determine whether the reaction is endothermic or exothermic [29]. Endothermic or exothermic reactions involve both physisorption and chemisorption sorption processes. These reactions indicate strong interaction between As(III)/As(V) and iowaite. The interactions with charged molecules were mostly endothermic and purely entropy-driven, indicating that the sorption on iowaite surfaces could be described as an interaction between opposite charges. In particular, the increase in entropy resulted from the release of surface-structured water molecules, and counterions from the electronic double layer supplied the major contribution to the free energy of sorption. Understanding the thermodynamic interactions of iowaite with arsenic is important to determine the forces behind their self-organization and co-organization with other compounds.

### 4.5. Adsorption of Different Forms of Arsenic

Due to the effect of salt in biological extract, the peak position of different arsenic was different from that of the standard. A separate standard solution was added to confirm that all the peaks numbered in Figure 8B,C were the same composition as those in Figure 8A. The order of the peaks in Figure 8A was as follows: AsB, As^3+^, DMA, MMA, and As^5+^. This result showed that the content of As^3+^ and As^5+^ decreased significantly after sorption, which was consistent with the above experimental results. The first peak was halved and a pre-peak appeared, which was considered AsC (Not in the mixed standard). The halved part of peak 1 and the sorption of peak 5 were the free state of As^3+^ and As^5+^. The sorption capacity to organic arsenic is limited, and the unknown form of arsenic or AsB may be transformed to increase the content of the third peak. The toxicity of arsenic varies according to As(III) > As(V) > MMA> DMA> AsB> AsC. Total arsenic determination by ICP-MS revealed that more than 50% of arsenic was adsorptive, most of which was inorganic arsenic with high toxicity.

## 5. Discussion

Iowaite were successfully synthesized in this study as arsenic adsorbent through a chemical precipitation reaction verified by the result of XRD. Arsenic adsorption of ion is related to the contents of Fe and Mg other than its agitation speed and method of mixing. The values of *Q_e_* increased with the extension of exposure time, when the maximum adsorption is not reached. Increasing the agitation speed may lead to the enhancement of particle movement and decreased the iowaite particle size, which may increase the rate of arsenic sorption. According to the speciation of arsenate and arsenite in solution as an influence of pH [26], the major types of arsenate in the solutions reacting with iowaite was AsO_4_^3−^, and that of arsenite was AsO_2_^−^. All of these types were oxyanions that facilitated the reconstruction of the layered structure of iowaite by acting as intercalated anions. The reconstruction was completed in a few minutes and did not decrease the absorption ability of arsenic. The sorption of arsenic by acid-treated iowaite can be studied in subsequent experiments. When the reaction alert is acidic, the pH of the solution gradually increased with the reaction time, which may be caused by the ligand exchange between arsenic and iowaite on the adsorbent surface [31]. Surface structure is not stable under acidic condition, just like acid causes minerals to corrode. The mechanism of sorption is both a physical and chemical process that holds arsenic at the interface between the liquid and solid phases. The results of XRD explain the chemical process, new compounds formed after sorption. To distinguish the kinetics equation based on the concentration of a solution from the sorption capacity of solids, this second-order rate equation was represented, called pseudo-second-order. The pseudo-second-order equation has the following advantages: it does not have a problem of assigning effective sorption capacity; and the sorption capacity, rate constant of pseudo-second-order, and initial sorption rate can all be determined from the equation without knowing any parameter beforehand. The coefficient of determination, *R*^2^, was used to select the most appropriate kinetic model via previously published methods [32]. Thus, the pseudo-Langmuir-2 might be a good fitting model because of its *R*^2^ value at the concentration of 0.15–5 mg/L. Moreover, *R*^2^ decreased as the concentration increased.

A preliminary experiment was conducted to determine the optimal equilibrium time for sorption, although many factors affect sorption equilibrium. Under controlled temperature and the amount of material addition, both initial concentration and salinity can affect the equilibrium time. The initial concentration of NaAsO_2_ varied from 0.15 to 20 mg/L, the same as H_3_AsO_4_. The sorption equilibrium time increased with the larger initial concentration. The most important factor that weakens the sorption of arsenic compounds is the presence of high amounts of natural organic substances, such as phosphate compounds and silicates in water, which may reduce the sorption property of materials. The sorption kinetic results of arsenic removal in iowaite are shown in Figure 7 and Table 4. Regardless of As(III) or As(V), the residual arsenic concentration in the solution decreased with the increase in the reaction time, and the arsenic removal sorption amount of iowaite increased with time. At these sorption levels, the process of using iowaite for the removal and recovery of a heavy metal ion is potentially more economical than current process technology. Moreover, the sorption of heavy metal ions depends on their initial concentrations, temperature, and contact time. The results showed that iowaite resulted in good sorption on NaAsO_2_ and H_3_AsO_4_, both fitted well in the Langmuir model. The sorption kinetics of iowaite was fast (≤30 min), and it showed an improvement in performance with high temperature.

Numerous technologies have evolved for the removal of arsenic, out of which many are successful in the laboratory, but in practical application, they are not so effective. The efficiency of those materials decreases due to competition from other natural-occurring ions with arsenic for adsorbent sites. Iowaite performed well in inorganic arsenic removal from both standard solution and extraction from *M. veneriformis*. Simple experiments have been carried out to verify its zero absorption of nutrients, which may be due to its mineral origin. It can be used as a purification kit for live aquatic products and seafood containing arsenic with minimal modification at the later stage of preparation. The use of a purification kit can reduce the frequency of water change and prevent the secondary pollution of purified water. The sorption tests showed that the process is efficient even at high arsenic concentrations. The rapid removal process and good removal results make iowaite attractive for in situ and commercial (filters) use, because time and efficiency are essential requirements in new technologies. More other types of seafood with high arsenic content were needed to verify the universality of iowaite. A more systematic test on the stability and sorption of iowaite at wide ranges of pH and arsenic concentration needs to be performed before it will be applied in acid conditions. The dearsenication capacity may not be the greatest among some common sorbents reported in the literature, but the advantages in terms of acquisition costs and the introduction of other substances are remarkable. It provides an effective way for the industrial removal of products. Therefore, cheap and efficient techniques are essential to replace traditional treatment methods. Treating water and oceans containing hazardous arsenic by using a simple and inexpensive method is essential for human healthcare and environment. The major disadvantage of existing adsorbents is that they produce a huge amount of secondary pollutants with a considerable concentration of arsenic. The management of the contaminated adsorbent is important for safeguarding the environment from secondary pollution. Iowaite is a mineral of natural origin, which can be disposed of ores and does not absorb nutrients; after a long period of mineralization, it may become available resources. On the other hand, iowaite can be totally degraded by different acids, which is likely to be converted to industrial Mg or Fe compounds through other synthesis steps.

## Figures and Tables

**Figure 1 molecules-26-03052-f001:**
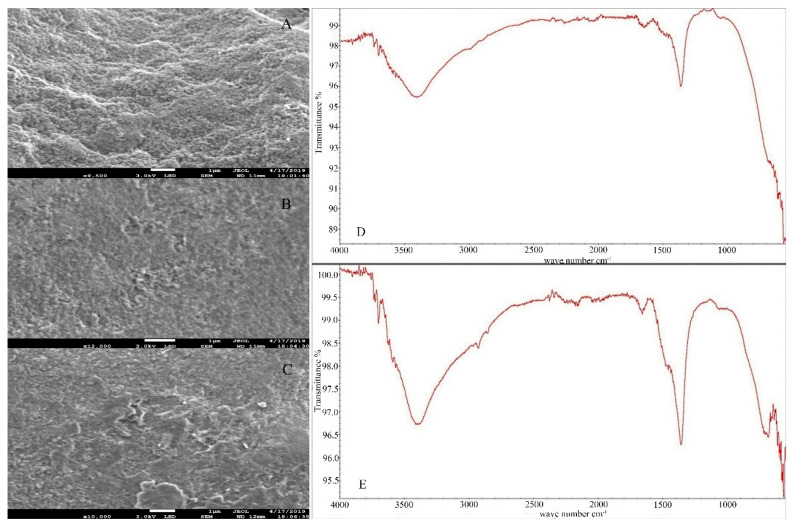
SEM images under 1 µm (**A**): iowaite, (**B**): iowaite-NaAsO_2_, (**C**): iowaite-H_3_AsO_4_ FT-IR spectra (**D**): iowaite-As; (**E**): iowaite.

**Figure 2 molecules-26-03052-f002:**
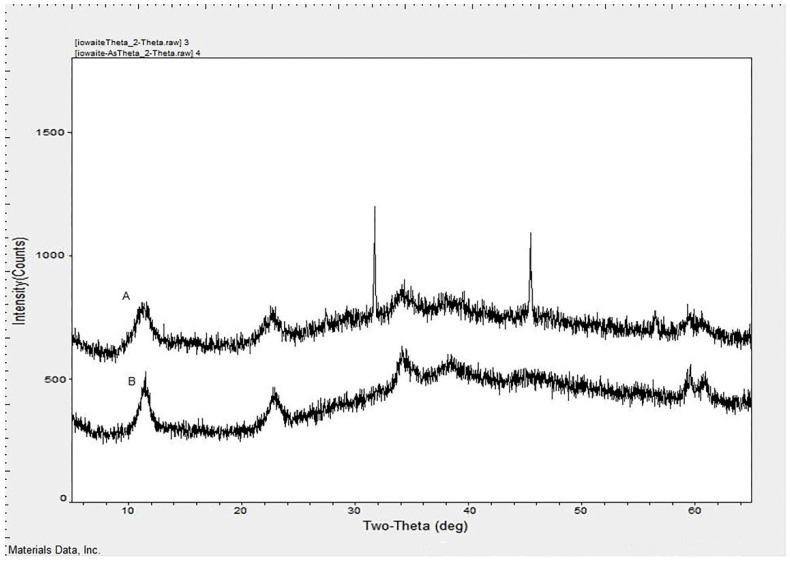
PXRD patterns of iowaite (**A**) and iowaite-As (**B**).

**Figure 3 molecules-26-03052-f003:**
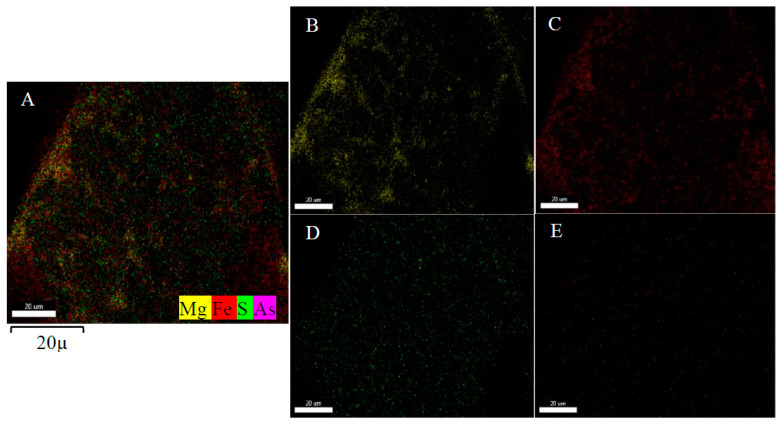
Iowaite-NaAsO_2_ EDS images of each element channel ((**A**) covers channels of Mg, Fe, S and As, (**B**) is the channel of Mg, (**C**) is the channel of Fe, (**D**) is the channel of S, (**E**) is the channel of As).

**Figure 4 molecules-26-03052-f004:**
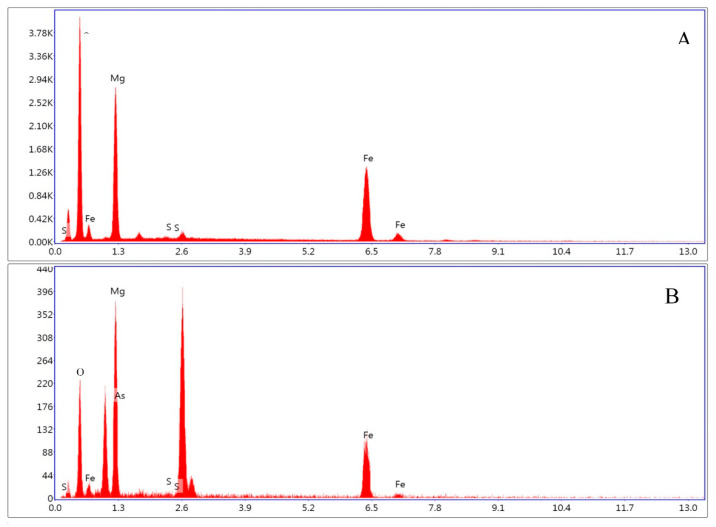
Line scan image of EDS (iowaite (**A**) and iowaite-NaAsO_2_ (**B**)).

**Figure 5 molecules-26-03052-f005:**
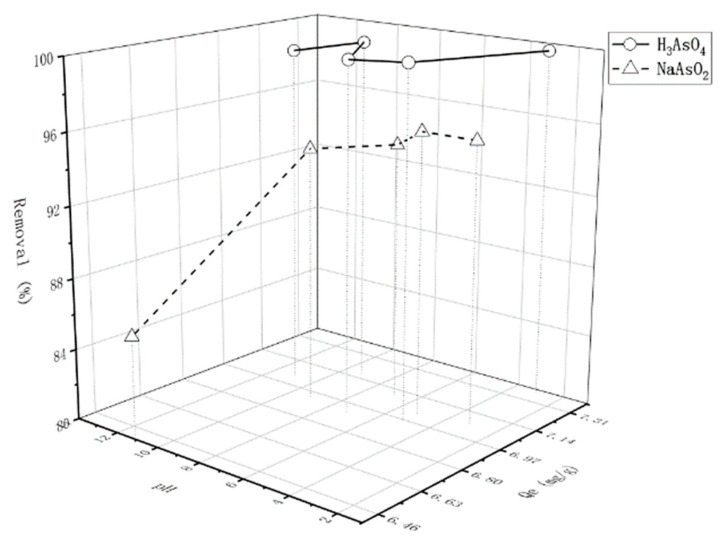
Effect of pH on arsenic removal by iowaite (-○-: H_3_AsO_4_; -Δ-: NaAsO_2_).

**Figure 6 molecules-26-03052-f006:**
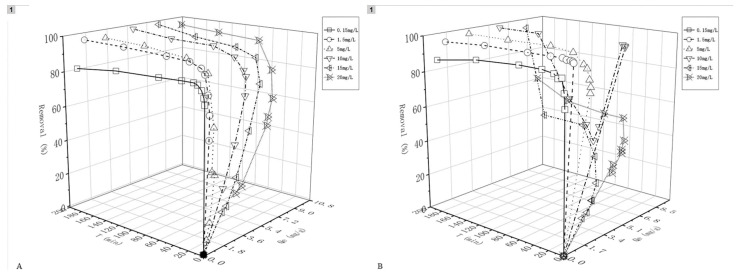
Qe and removal of different concentrations with time ((**A**): NaAsO_2_, (**B**): H_3_AsO_4_).

**Figure 7 molecules-26-03052-f007:**
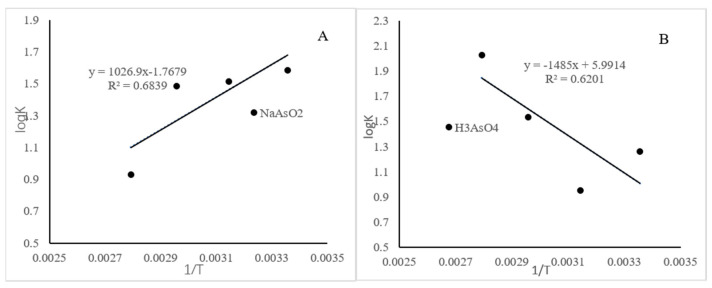
Regression curve of Langmuir model ((**A**): NaAsO_2_; (**B**): H_3_AsO_4_).

**Figure 8 molecules-26-03052-f008:**
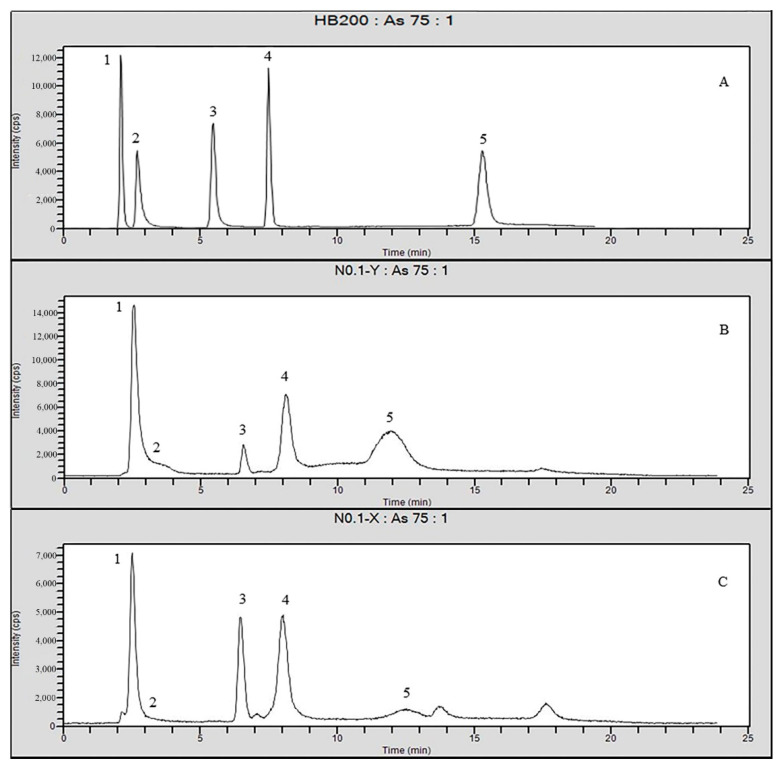
Chromatograms of HPLC-ICP-MS ((**A**): Mixed standards, (**B**): Extract of *Mactra veneriformis,* (**C**): iowaite disposal of the *Mactra veneriformis* products; (Peak 1-AsB; Peak 2-As^3+^; Peak 3-DMA; Peak 4-MMA; Peak 5-As^5+^).

**Table 1 molecules-26-03052-t001:** Instrumental operating condition for HPLC-ICP-MS.

**ICP-MS Parameters**	
Plasma power	1650 W
Nebulizer gas flow	0.85 L/min
Auxiliary gas flow	1.2 L/min
Plasma gas flow	18 L/min
Monitored ion	m/z 75 (^75^As)
Reaction Mode	KDC
Dwell Time	250 ms
Acquisition Rate	4 pt/s
**HPLC Parameters**	
Column	Hamilton PRX-X100 (250 mm × 4.1 mm, 10 μm)
Column temperature	30 °C
Mobile phase A	25 mmol/L NH_4_HCO_3_ adjusted to pH 8 with NH_4_OH
Mobile phase B	Water
Gradient program	0–15 min: 0–100% A; 16–19 min: 100–0% A; 20–24 min: 0% A
Flow rate	1 mL/min
Injected volume	50 µL

**Table 2 molecules-26-03052-t002:** Effect of pH on H_3_AsO_4_ removal by iowaite.

Initial pH	Equilibrium Concentration/(mg/L)	Dosage of Adsorbent/mg	Q_e_/(mg/g)	Removal/%
3.01	0.0046	10.14	7.37	99.69%
5.13	0.0039	10.74	6.96	99.74%
7.14	0.0047	10.82	6.91	99.69%
9.17	0.0051	10.43	7.17	99.66%
11.08	0.0121	10.57	7.04	99.19%

**Table 3 molecules-26-03052-t003:** Effect of pH on NaAsO_2_ removal by iowaite.

Initial pH	Equilibrium Concentration/(mg/L)	Dosage of Adsorbent/mg	Q_e_/(mg/g)	Removal/%
3.13	0.0636	10.17	7.06	95.76%
5.05	0.0607	10.25	7.02	95.95%
6.80	0.0807	10.01	7.09	94.62%
9.08	0.0835	10.22	6.93	94.43%
11.24	0.2154	10.02	6.41	85.64%

**Table 4 molecules-26-03052-t004:** Parameter of pseudo-first-order and pseudo-second-order models.

Kinetic Parameter	*Co*/(mg/L)	Pseudo-First-Order Model			Pseudo-Second-Order Model		
		*k* _1_	*q_e_*	*R* ^2^	*k* _2_	*q_e_*	*R* ^2^
NaAsO_2_	0.150	1.922	0.065	0.991	82.036	0.066	0.999
	1.500	0.897	0.683	0.976	2.196	0.710	0.998
	5.000	0.106	2.370	0.977	0.058	2.574	0.986
	10.000	0.428	4.597	0.987	0.128	4.880	0.997
	15.000	0.132	6.951	0.980	0.026	7.492	0.985
	20.000	0.212	8.741	0.936	0.032	9.431	0.986
H_3_AsO_4_	0.150	1.612	0.065	0.994	62.743	0.067	0.999
	1.500	4.131	0.707	1.000	72.370	0.708	0.999
	5.000	1.706	2.399	0.976	1.611	2.457	0.992
	10.000	196.645	4.372	0.768	−	4.372	0.768
	15.000	0.023	5.872	0.830	−	3.920	0.242
	20.000	0.527	6.856	0.817	1.949	6.142	0.598

**Table 5 molecules-26-03052-t005:** Isothermal sorption parameters of iowaite.

Solution	*T*/°C	Freundlich Model			Langmuir Model		
		1/*n*	*k* _4_	*R* ^2^	*Q_m_*	*k* _3_	*R* ^2^
NaAsO_2_	25	0.268	0.073	0.996	2.17	38.38	0.996
	45	0.275	0.069	0.976	2.45	32.47	0.996
	65	0.342	0.117	0.891	1.81	30.48	0.946
	85	0.374	0.108	0.944	8.62	8.53	0.999
H_3_AsO_4_	25	0.068	2.78	0.468	5.20	18.12	0.929
	45	0.177	2.87	0.594	1.99	8.92	0.981
	65	0.486	146.96	0.951	1.20	34.30	0.982
	85	0.294	20.85	0.784	3.66	21.49	0.571

**Table 6 molecules-26-03052-t006:** Adsorption thermodynamic parameter of iowaite.

	*T*/K	*k* _3_	Δ*G*^0^ (J/mol)	Δ*H*^0^ (J/mol)	Δ*S*^0^ (J/mol)
NaAsO_2_	298	38.38	−9.04	−8.54	−14.70
	318	32.47	−9.20		
	338	30.48	−9.60		
	358	8.53	−6.38		
H_3_AsO_4_	298	18.12	−7.18	12.68	49.81
	318	8.92	−5.79		
	338	34.30	−9.93		
	358	21.49	−13.89		

## Data Availability

The data presented in this study are available on request from the corresponding author.

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
