# Peer review of "Synthetic Iowaite Can Effectively Remove Inorganic Arsenic from Marine Extract"

_molecules, 2021, doi:10.3390/molecules26103052_

Round 1

Reviewer 1 Report

In my opinion, the revised manuscript is still moderate in significance and impact.  
However, the authors did their maximum (to the best of their knowledge) to improve the manuscript.
From this standpoint, the manuscript might be considered for the next step.  

Specific comment (if any): 
The entropy (in Table 6) should be reported in the (J/mol) unit not in (kJ/mol), please check the calculations. 

Author Response

Cover Letter

Dear Editors:

We would like to revise the enclosed manuscript entitled “Synthetic iowaite can effectively remove inorganic arsenic from marine extract”, which we wish to be considered for publication in the journal of Molecules.

Responses to the reviewers' comments all listed.

For the first review:

  1. The entropy (in Table 6) checked to be (J/mol) unit not in (kJ/mol). 

For the second review

  1. Topic 2.4 and 2.5 was adjusted to 2.1
  2. Characteristic peak of FTIR and EDS were added in topic 4.1. The order of the pictures has been adjusted for better comprehension of the lectors.
  3. The reader will often be attracted by the direct information of the pictures as he goes through the whole article. Fig 5 was necessary.
  4. Different size of font in page 9 and 11 were all revised.

Sincerely yours,

Jianming Cheng

School of Pharmacy

Nanjing University of Chinese Medicine

138 Xianlin Avenue, Nanjing, 210023, China

Phone: +8613951835384

Reviewer 2 Report

Dear authors,

The paper entitled " Synthetic iowaite can effectively remove inorganic arsenic from 1 marine extract" is really relevant for the area but needs a minor revision for publication in Molecules journal. The results and discussion is very appropriated and with a great application.

Below I listed some comments about text revision:

  1. Some adjusts in mentioning the methodology are need, such as including the detailed information about equipment and conditions used in characterization in the topic 2.4 and not inside the presented results.
  2. The results presented about characterization of iowaite and after adsorption iowaite may be very improved, in relation of the detailed information showed. The quality of images of Figures 1, 2, 3 and 4 is very poor and the detailed citation of the figures in the text must change for better comprehension of the lectors. Inside the images of FTIR and EDS the authors can show clearer the characteristic peak, the element in a correct local.
  3. The authors must improve the figures 5 and 8 and decided if Figure 5 is really necessary, due to the Tables 2 and 3 show with detail they results obtained.
  4. About form, in a final of page 9 it is possible to see different size of font used in the text.

Author Response

Cover Letter

Dear Editors:

We would like to revise the enclosed manuscript entitled “Synthetic iowaite can effectively remove inorganic arsenic from marine extract”, which we wish to be considered for publication in the journal of Molecules.

Responses to the reviewers' comments all listed.

For the first review:

  1. The entropy (in Table 6) checked to be (J/mol) unit not in (kJ/mol). 

For the second review

  1. Topic 2.4 and 2.5 was adjusted to 2.1
  2. Characteristic peak of FTIR and EDS were added in topic 4.1. The order of the pictures has been adjusted for better comprehension of the lectors.
  3. The reader will often be attracted by the direct information of the pictures as he goes through the whole article. Fig 5 was necessary.
  4. Different size of font in page 9 and 11 were all revised.

Sincerely yours,

Jianming Cheng

School of Pharmacy

Nanjing University of Chinese Medicine

138 Xianlin Avenue, Nanjing, 210023, China

Phone: +8613951835384

This manuscript is a resubmission of an earlier submission. The following is a list of the peer review reports and author responses from that submission.

Round 1

Reviewer 1 Report

The paper may be published after some revisions:

  1. English language must be improved.
  2. All figures are not so clear, the scales on the axes must be enlarged.
  3. In the Material and Methods section, it is necessary write some explanation sentences about reagents, only the table is not sufficient.
  4. At row 94, abbreviation LDHs must be defined.
  5. The section 3.3 is not clear, a description of this adsorption is needed, and the argument must be
  6. FESEM analysis is not so clear, some sentences must be added.
  7. At line 166, about XRD spectrum, two peaks are mentioned but what they are must be explained.
  8. At row 250, absorption must be changed with adsorption.
  9. Conclusions and discussion must be separated in two sections. Conclusions must be only a summarization of the work. The discussion should be included in the results section.

Reviewer 2 Report

This manuscript (MS Ref No: molecules-993792) is concerned with the preparation and characterization of synthetic iowaite (MgO/Fe2O3) and its application for adsorption of arsenic ionic-species (As(III) & As(V)) from aqueous solutions.  After evaluation, it seems that this particular manuscript is not very attractive in terms of novelty, convincing style, and implications. The quality of the manuscript is below average compared to the high standard of the Journal.

From my standpoint, the reported research contains some errors and inappropriate interpretations. For example, from the comments about XRD (Fig.2) the authors did not confirm the patterns of Iowite, the comments are superficial. Did the peaks correspond to a specific CARD number? On page 5 line 165 – what did you mean by “..it was an amorphous polymer”?; do you mean – an amorphous state of the inorganic phase? The adsorption capacity Qe(mg/g) reported in Tables 2 and 3 are below 1 mg/g, which are not of practical interest. In this sense, proper adsorbents should bestow sorption capacity at least greater than 10 mg/g. The parameter k2 in Table 4 has two negative values -1.544 and -1.101E+45?, which violate its physical meaning. I suppose this is the consequence of the inappropriate regression technique. The parameters were fitted based on linear regression only.  Now, modern nonlinear regression tools are widely available (in Excel, LibreOffice Calc, QtiPlot, Origin, SigmaPlot, Minitab, Scilab, etc.). Hence, non-linear regression should be applied to diminish the mathematical errors of the fitting.  The plots of kinetics and isotherms are missing.